# Use of Bimodal Particle Size Distribution in Selective Laser Melting of 316L Stainless Steel

**Hannah G. Coe** [1,2] **and Somayeh Pasebani** [1,2,*]

1   School of Mechanical, Industrial and Manufacturing Engineering, Oregon State University, Corvallis, OR 97330, USA; h.coe73@gmail.com
2   Advanced Technology and Manufacturing Institute (ATAMI), Corvallis, OR 97330, USA
*   Correspondence: somayeh.pasebani@oregonstate.edu; Tel.: +1-541-737-3685

**Abstract:** Spherical powders with single-mode ($D_{50}$ = 36.31 μm), and bimodal ($D_{50,L}$ = 36.31 μm, $D_{50,s}$ = 5.52 μm) particle size distribution were used in selective laser melting of 316L stainless steel in nitrogen atmosphere at volumetric energy densities ranging from 35.7–116.0 J/mm$^3$. Bimodal particle size distribution could provide up to 2% greater tap density than single-mode powder. For low laser power (107–178 W), where relative density was <99%, bimodal feedstock resulted in higher density than single-mode feedstock. However, at higher power (>203 W), the density of bimodal-fed components decreased as the energy density increased due to vaporizing of the fine powder in bimodal distributions. Size of intergranular cell regions did not appear to vary significantly between single-mode and bimodal specimens (0.394–0.531 μm$^2$ at 81–116 J/mm$^3$). Despite higher packing densities in powder feedstock with bimodal particle size distribution, the results of this study suggest that differences in conduction melting and vaporization points between the two primary particle sizes would limit the maximum achievable density of additively manufactured components produced from bimodal powder size distribution.

**Keywords:** 316L; selective laser melting; additive manufacturing; bimodal powder; particle size distribution

## 1. Introduction

Selective laser melting (SLM) is a laser powder bed fusion (LPBF) additive manufacturing (AM) process in which three-dimensional parts are manufactured by scanning a laser in a prescribed pattern on the surface of a bed of metal powder, melting the material which rapidly solidifies, before a new layer of powder is spread to repeat the process. UNS S31603, known as 316L stainless steel, is a low-carbon, austenitic steel, with a standard material density, $\rho_m$, of approximately 8.0 g/cm$^3$ [1]. The AISI 316 L is the most commonly used stainless steel in industry due to its strength, toughness, and corrosion resistance [2] and, therefore, 316L is a desirable system for use in SLM and this study.

Mechanical properties of 316L steel, even at high temperatures, were shown in some studies to be improved in SLM-manufactured 316L steel compared to the wrought material [3–6]. In a study by Bartolomeu et al. [5], SLM-manufactured components were found to have up to 41% higher yield strength and as much as 144% higher tensile strength than cast 316L. However, Suryawanshi et al. [4] found that while yield strength was also up to 144% higher than conventional machining, SLM only provided a marginal improvement of up to 28% in ultimate tensile strength. The high strength of SLM manufactured components is attributed to the refined grain size that develops due to rapid heating and cooling rates in SLM, resulting in a high energy required to move dislocations across grains [5].

Many of the desirable properties of 316L are attributed to its fully austenitic phase [3]. Due to the low carbon content in 316L (<0.03 wt.% carbon where wt.% is weight%) and high thermal

gradients in SLM, material cools quickly and remains well within the austenite regime after heating. Previous studies [7,8] used x-ray diffraction (XRD) to demonstrate that precursor powder feedstock maintains a near complete austenite phase before and after printing. While the dominant phase in both gas-atomized powder and SLM-processed 316L is austenite ($\gamma$), slight secondary ferrite ($\delta$) peaks have been demonstrated for powder feedstock, likely due to phase changes during solidification in the powder atomization process [7,9].

The intensity of a laser beam as it melts metal powder during SLM is dependent on parameters such as power (*P*) [W], scan speed (*v*) [mm/s], layer thickness (*t*) [μm], and laser beam diameter (*σ*) [μm]. These parameters can be combined into a single metric of establishing the amount of energy going into the powder, known as the volumetric energy density (VED) that can be calculated according to Equation (1) [10,11], though in some studies, σ is defined by hatch spacing, or the distance between the center of two parallel melt pools in an SLM scanning pattern [12,13].

$$VED = \frac{P}{v\sigma t} \left[ \frac{J}{mm^3} \right] \tag{1}$$

According to Bertoli et al. [10], VED does not provide a complete picture for predicting melt pool behavior or component density, as it fails to account for complex melt pool physics. For example, Marangoni flow and recoil pressure can influence the continuity of the melt track that are not well captured in the above VED equation [14]. Marangoni flow is a mass flow driven by surface tension gradients in a fluid [15]. In a melt pool during SLM, high temperature gradients can produce differences in surface tension within the liquid metal, causing this kind of flow to occur along the boundaries of the melt pool [16]. Furthermore, recoil pressure is caused by the rapid expansion of metallic gas when a laser heats the material beyond its vaporization temperature during SLM [16]. The effect of these processes can differ for various laser powers and scan speeds, even if those parameters equate to the same VED number. While VED can be used to help predict the continuity of a single melt track, other parameters must be considered to produce high-density SLM-manufactured parts and to ensure of achieving a full, uniform melting and sufficient overlap between tracks. These parameters include hatch spacing, shift angle, build orientation, and gas atmosphere [4,7,9,17,18]. Thus far, most efforts to increase the density of SLM-manufactured components have focused on the process parameters and machine settings [9,12,14,16,19,20] rather than the density of the powder bed itself.

In SLM, feedstock typically consists of spherical powder with a single-mode normal size distribution [21]. However, early work by McGeary [22], later elaborated upon by German [23], has reported on the benefit of bimodal size distributions to achieve higher powder bed packing densities than powders with a single primary particle size. This concept has enormous potential in powder based additive manufacturing processes, where maximizing powder bed density is crucial to maximizing final component density. McGeary [22] found that for these bimodal powders, a size ratio between large and small particle size of approximately 1:7, and a composition of approximately 30% fine particles can produce optimal packing density. In addition to experimental validation, the size relationship is expressed geometrically according to Equation (2), where *r* is the radius of fine particles, and *R* is the radius of coarse particles.

$$r = \left( \frac{2}{\sqrt{3}} - 1 \right) \times R \equiv \frac{1}{7}R \tag{2}$$

The weight percentage of fine powders, $X_{fines}$, that is required for optimal packing density is determined from the relative density of the fine and coarse powders, according to Equation (3). Theoretically, assuming 60% for relative density of both the coarse powder, $\rho_{coarse}$, and the fine

powder, $\rho_{fines}$, a maximum density, $\rho^* = 0.84$ times the material density can be achieved using $X_{fines} = 30$ wt.% fines.

$$X_{fines} = 1 - \frac{\rho_{coarse}}{\rho^*} = 1 - \frac{\rho_{coarse}}{\rho_{coarse} + (1 - \rho_{coarse}) \times \rho_{fines}} \tag{3}$$

Karapatis et al. [24] used this work in a study on the effect of particle size distribution (PSD) in selective laser sintering (SLS), using smooth, spherical particles of nickel-based alloy, with sizes ranging from 20 to 200 µm in diameter. This study found that for high ratios of small to large particle size (over 1:10), and a mixing ratio of 30 wt.% fine powder, the relative apparent density of the bimodal powder was up to 15% higher than that of single-mode powder ($AD_{bimodal} \approx 63\%$, $AD_{single} \approx 55\%$). Do et al. [25] utilized McGeary's model [22] for measuring and mixing bimodal powders in binder jetting and found that use of bimodal powder not only increased the packing density of the powder bed, but also improved the surface finish, and reduced the distortion and shrinkage of parts that were associated with the sintering step [25]. Similarly, Zhu et al. [26] utilized this concept to perform direct laser sintering of a bimodal mixture of dissimilar metal powders (60 wt.% spherical Cu powder with a 31–88 µm diameter combined with 40 wt.% spherical SCuP with a size range of 5–20 µm), and found a notably higher density in samples sintered from powder mixtures with higher apparent densities.

Despite these studies, the potential for using bimodal feedstock of a uniform alloy composition undergoing full melting in SLM has been largely unexplored [27]. The objective of this study is to evaluate the use of spherical 316L stainless steel powder bed feedstock with both single-mode PSD (typically used in the SLM process) and bimodal PSD and compare density, mechanical properties and the microstructure after SLM and after SLM followed by annealing.

## 2. Materials and Methods

The 316L stainless steel powder feedstock used in this research included single-mode, gas-atomized 316L powder procured from GKN Hoeganaes (GH), with a reported particle size of $D_{50} = 36.31 \pm 11.92$ µm; and a smaller, semi-spherical water-atomized and gas-atomized powder from US Research Nanomaterials (USRN) with a primary particle size of $D_{50} = 5.52 \pm 2.5$ µm. Table 1 gives the nominal chemical composition reported by each manufacturer, along with the industrial standard composition ranges for 316L determined by the American Society for Metals (ASM) [1].

**Table 1.** Composition of powders reported by the vendor (wt.%).

| Supplier | Fe | Cr | Ni | Mo | Si | Mn | C | O | S | N |
|----------|-----|-------|-------|-----|------|------|-------|-------|-------|-------|
| GH | | 16.6 | 10.7 | 2.4 | 0.35 | 1.5 | 0.02 | 0.06 | 0.007 | 0.06 |
| USRN | Bal. | 16–18 | 12–15 | 2–3 | <1 | <0.3 | <0.03 | <0.15 | - | - |
| ASM [1] | | 16–18 | 10–14 | 2–3 | <0.75 | <2 | <0.03 | - | <0.03 | <0.10 |

### 2.1. Powder Characterization and Processing

Apparent density (AD) of the powders was measured by pouring powder through a standard Hall flow cone [28] into a container with volume of 25 cm$^3$ and measuring the mass of the volume after leveling. Apparent density of a bulk powder is often similar to the density of a spread powder bed [26]. Tap density (TD) was measured according to standard procedure ASTMB527 [29] using a Quantachrome Autotap AT-6-110-60 mechanical tapper with a 100 cm$^3$ graduated cylinder.

An attribute used in this study to determine good flowability of a powder is the Hausner ratio, H. As early as 1969, the Hausner ratio has been observed to have a relationship to properties of metal powders [30,31]. The Hausner ratio, or the ratio of tap density ($\rho_t$) to apparent density ($\rho_a$), was calculated according to Equation (4), and used as an indicator to compare flow properties of



each powder type. Powders with a Hausner ratio less than 1.25 are generally considered to be free flowing [26].

$$H = \frac{\rho_t}{\rho_a} \tag{4}$$

Bimodal powder was created by first calculating the optimal mass fraction of large and small particle types according to Equation (3), using the tap density of each powder. Once the weight of each powder was measured, the bimodally distributed powder was mixed first by manually inverting the container for 50+ times, then mixing with a vortex mixer for 30 s at ~3000 rpm. This was done in batches of ~500 cm$^3$ each. Then the batches were combined, followed by additional 20+ inversions. Once mixed, the AD and TD of the bimodal powder was measured using the previous methods. An FEI Quanta 600F environmental scanning electron microscopy (SEM) was used to observe the morphology of the small, large, and bimodal powders. Each SEM powder sample was prepared by applying a small amount of powder to carbon tape adhesive on the sample's holder, then using sweeping away excess loose powder using a hose of flowing nitrogen gas.

*2.2. Selective Laser Melting Processing Parameters*

SLM was performed using an OR LASER CREATOR SLM machine with a 1070 nm Yb Fiber Laser with 250 W power under nitrogen atmosphere. Features of the build chamber in the Creator are shown in Figure 1.

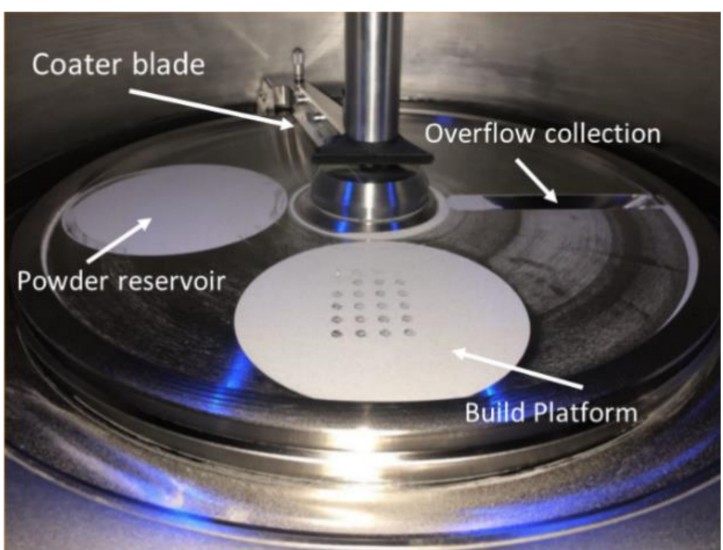

**Figure 1.** Build chamber architecture in ORLAS CREATOR selective laser melting (SLM) machine.

The system uses a rotational spreading method to feed powder from the reservoir, across the build platform with diameter of 10 cm, and into the overflow collection chamber. The height-adjustable coater arm is equipped with a replaceable rubber blade, to evenly spread powder across the build platform and ensure minimal wear on mechanical components. Oxygen content in the system was maintained below 0.1 volume% (vol%). Samples were produced for GH single-mode powder and the bimodal powder mixture for volumetric energy densities ranging from 35.7 to 116.0 J/mm$^3$ as shown with scan speed, power and constant parameters in Table 2. For each powder, 5 mm × 5 mm ($\phi$ × height) cylindrical specimens were produced on a single build plate with a single sample for each VED, spaced 5 mm apart.

Samples were built with support structures to ensure consistent layer spreading in the bulk of the sample, as well as to prevent warping of the sample geometry during melting. The supports were designed as 1 cm tall pillars with a cross-shaped cross-section. The total width of each cross-section was 1.5 mm in each direction with a thickness of 0.5 mm, and each pillar was spaced in a grid pattern with 0.5 mm between each from edge to edge. Samples were removed from the build plate with support

structures still attached. This was done manually using a hammer and chisel to gently tap at the base of the support structure near its fusion to the build plate. After removal from build plate, support structures were removed from the bottom of each sample using a rotary diamond saw at 600 rpm.

**Table 2.** Parameters matrix used for optimization of the SLM process (power and scan speed were variables while beam dimeter (50 μm), hatch spacing (50 μm), layer height (50 μm), and shift angle (45°) were unchanged).

| VED (J/mm$^3$) | | Power (W) | | | |
|:---:|:---:|:---:|:---:|:---:|:---:|
| | | 107 | 139 | 178 | 203 |
| | 700 | 61.1 | 79.4 | 101.7 | 116.0 |
| | 800 | 53.5 | 69.5 | 89.0 | 101.5 |
| Scan Speed (mm/s) | 900 | 47.6 | 61.8 | 79.1 | 90.2 |
| | 1000 | 42.8 | 55.6 | 71.2 | 81.2 |
| | 1100 | 38.9 | 50.5 | 64.7 | 73.8 |
| | 1200 | 35.7 | 46.3 | 59.3 | 67.7 |

Microhardness tests were conducted on the cut and polished surface of SLM-manufactured samples made from GH and BIMO powder at different VED values ranging from 66 to 116 J/mm$^3$. Indentations were made in 10 random locations around the cut and polished surface perpendicular to the build direction for each sample using 500 g force and 13 s dwell time.

*2.3. Post-Processing and Characterization*

The density of each sample was measured using the Archimedes method. An OHAUSE PA84 digital balance equipped with an Archimedes density testing kit was used with deionized (DI) water. To ensure all powder and SLM-manufactured component relative densities were an accurate reflection of the parent material, bulk density was measured for each precursor powder by first Arc Melting an SLM-manufactured sample produced from each single-mode and bimodal powder batch so that a full-density sample of each powder type was represented. Then, the density of these melted samples was measured using the Archimedes method. Relative powder density and SLM sample densities shown in the results of this report are based on parent material densities recorded from these measurements.

Microstructure analysis was carried out by cutting along planes normal and parallel to the build direction using a Pace Technologies PICO 155P precision cutter with circular diamond coated blade and at 600 rpm. Samples were then hot mounted at 370 °C using a Pace Technologies TP-7001B pneumatic mounting press. Mounted samples were then finished with 240, 400, 800, 1200 SiC grinding papers using a Pace Technologies NANO-2000T manual polisher at 150 rpm with a FEMTO-1100 rotating concurrently at 100 rpm. Then samples were polished by diamond suspension of 3, and 1 μm at 100 rpm. To reveal the microstructure, polished samples were then electro-etched in solution of 10% oxalic acid at 15 V for 15 s.

The phases present in both powder and bulk SLM samples were identified by X-ray Diffraction (XRD) using a Bruker D8 X-ray diffractometer. Continuous scanning mode was used to conduct scans at 2°/min with a dwell time of 1.5 s for a range of 2θ = 20–90° at step size of 0.05°. Optical microscopy and SEM imaging were used to observe the porosity, melt pool and microstructure of SLM-manufactured samples produced from single-mode (GH) and bimodal (BIMO) powders. Using Image J software, the area fraction of pores was quantified and compared. SEM images of microstructure and finer porosity features were obtained using the same Quanta 600 machine used in powder imaging. Selected samples were annealed by heating at a ramp rate of 8.5 °C/min and holding for 2 h at 1020 °C under a nitrogen atmosphere to ensure a fully austenitic phase was achieved. After annealing, samples were cooled in the furnace under nitrogen, then polished and electroetched using methods described above. The hardness of each sample was measured using a Leco LM 248AT Vickers microhardness tester with three samples taken from the normal and build plane of each specimen. Loading was at done at 500 g with a dwell time of 13 s. For each powder type and parameter set measured, 10 indentations were made to record the average hardness and standard deviation.

## 3. Results and Discussion

### 3.1. Powder

Figure 2 shows SEM images of the small (USRN) and large (GH) powders used in the bimodal mixture, BIMO. The USRN powder used here showed morphology that was spherical with minimal fused and satellite particles, contributing to notably improved flow and packing indicators, as shown in Figure 3. With an average size ratio of 6.6:1, the two powders that compose BIMO closely matched the 7:1 size ratio as determined by McGeary [22]. Figure 3a presents a comparison of TD and AD values for each of the single-mode and bimodal powders used in producing SLM-manufactured samples. The maximum packing density of the bimodal ($5.23 \pm 0.11$ g/cm$^3$) was slightly higher than that of the single-mode powder ($TD_{GH} = 5.09 \pm 0.1$ g/cm$^3$). Small powders and bimodal powders containing the smalls had a significantly lower AD than the larger single-mode powders. For example, AD of GH was $4.63 \pm 0.01$ g/cm$^3$, whereas AD of BIMO was $4.28 \pm 0.08$ g/cm$^3$, and AD of the USRN fine powder was as low as $3.41 \pm 0.13$ g/cm$^3$.

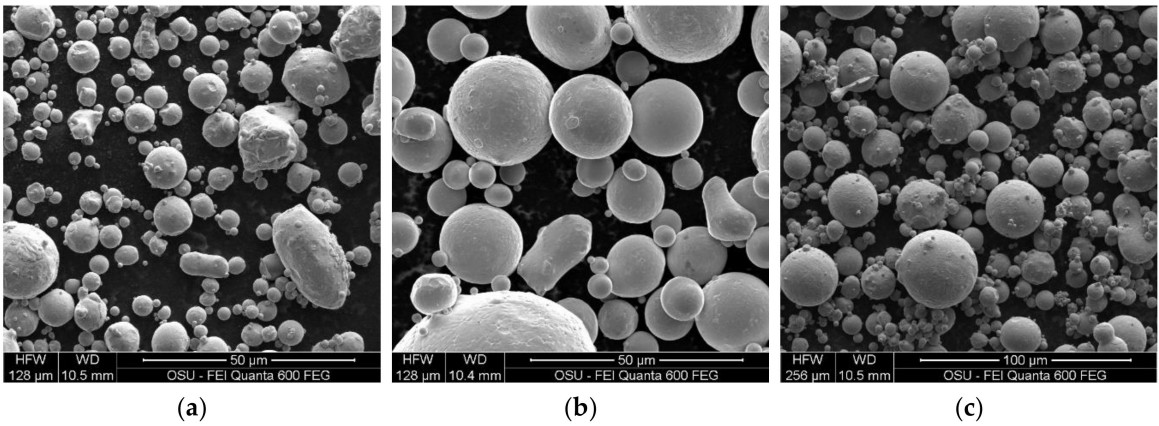

(a)    (b)    (c)

**Figure 2.** Scanning electron microscopy (SEM) micrographs of the powders used in the bimodal mixture, including (**a**) the smaller, USRN; (**b**) larger powder, GH; and (**c**) bimodally distributed powder, BIMO.

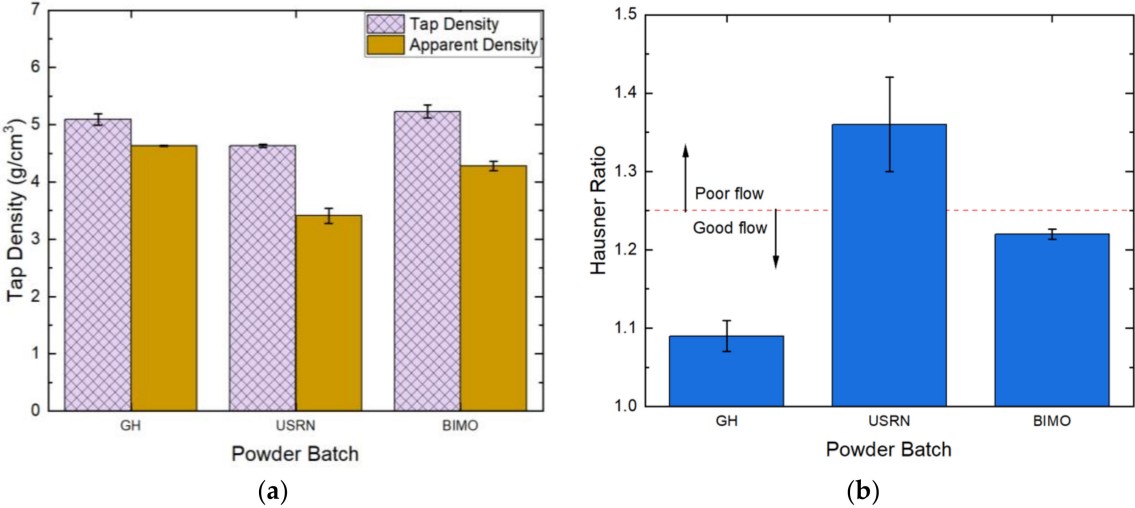

(a)    (b)

**Figure 3.** (**a**) Powder bed densities for each powder type; (**b**) Hausner ratio measured for each powder type. The red dotted line shows upper limit of good flow at H < 1.25.

The measured Hausner ratios of the fine powder (USRN) and bimodal powder (BIMO) were much higher than the threshold for good flow (H < 1.25), and thus were considered to have poor flowability. One reason for poor flowability in mixtures containing small particles was that particles

with higher surface area to mass ratios tended to absorb more moisture from the air. However, while mixed BIMO had poor flowability compared to the single-mode large powder (GH), the Hausner ratio for this bimodal powder was sufficiently low to classify the powder as having "good flow" according to the standard [30,31].

*3.2. Density*

Figure 4 presents the relative densities of SLM-manufactured samples produced at different VED levels and laser powers using powder feedstocks with single-mode and bimodal powder size distribution. The densities were measured using the Archimedes method and were compared to nominal densities ($\rho_{GH}$ = 7.934 ± 0.003 g/cm$^3$ and $\rho_{BIMO}$ = 7.935 ± 0.008 g/cm$^3$, determined from arc-melted samples) to determine percentage relative density for each SLM-manufactured sample. For single-mode powder (GH), the maximum average density was found at a VED of 67.7 J/mm$^3$ and for bimodal feedstock, peak density was found at 89.0 J/mm$^3$. With relatively high laser power (~203 W) and high scan speeds (1000–1200 mm/s) at these VEDs, relative densities were measured to be over 99.9 ± 0.2%.

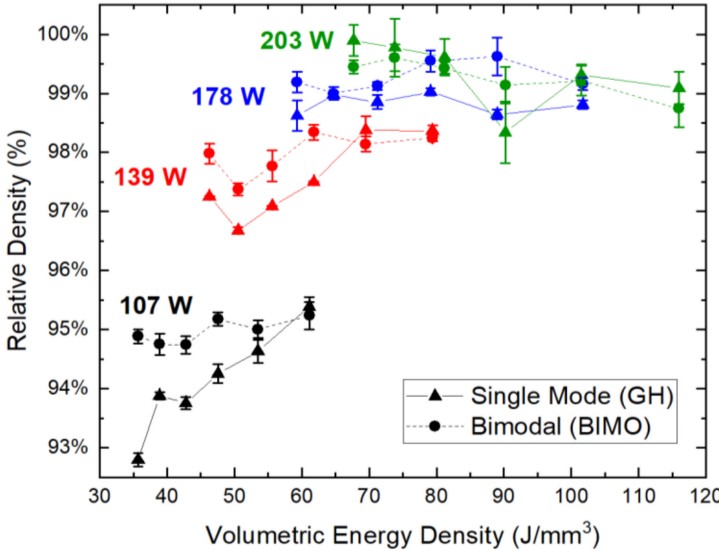

**Figure 4.** Relative density of 316L SLM-manufactured samples as a function of VED for single-mode (GH) and bimodal feedstock powder (BIMO), categorized by power level. For each power level, scan speed ranges from 700 to 1200 mm/s.

While the plot in Figure 4 shows an approximate trend towards increasing density with increasing VED values, the relationship is not consistent enough to achieve a curve fit that accurately describes this relationship with a mathematical equation. As implied by Bertoli et al. [10], using VED as a single parameter to predict density of an SLM-manufactured component ignores much metallurgical phenomena and process physics during the SLM process. However, a closer investigation of these same relative density against VED values separated by each level of power input reveals a trend between density and VED when power is held constant. For example, Figure 4 shows that for GH single-mode powder, density values increased with increasing VED at low power levels (P < 178 W, VED = 60–100 J/mm$^3$), but at a certain power threshold (~178 W), density remained relatively constant for increasing VED. At higher power levels (~200 W) for VED > 70 J/mm$^3$, density decreased with increasing VED. This is consistent with well-documented laser–material interactions which demonstrate keyhole melting and vaporization beyond a particular limit of VED [10,32].

When VED was increased beyond 90 J/mm$^3$, density values dropped for both single-mode and bimodal powder. However, similar VED values yielded different densities when performed at different power levels and confirmed Bertoli's results [10]. These VED values for maximum density contradict

the widely regarded assertation by Cherry et al. [19] that maximum density is achieved at an optimum VED of 125 J/mm$^3$ [19]. More recent studies [7,10,17] have found that Cherry's theory [19] is an over-simplification of the relationship between VED and relative component density which fails to consider limitations of VED as a single predictor for powder consolidation behavior.

At each power levels of 107, 139 and 178 W, the relative density values of bimodal powder were significantly higher than the density of single-mode powder. However, the density in Figure 4 of samples produced from bimodal powder (BIMO) behaved differently with increasing VED than the single-mode samples do. For bimodal samples, the relative density was nearly constant for increasing values of VED (i.e. decreasing scan speed) at a single power level. The limit at which density decreased with increasing VED occurs at a relative density of ~99.6% as opposed to the 99.9% density maximum of single-mode powder. This implies that vaporization may be occurring at higher power for bimodal powder, because the fine powder in each bimodal mixture is much smaller than the larger powder. Therefore, it is possible that VEDs high enough to achieve conduction melting for the larger particles may be well beyond the vaporization point of the small powders. This is likely why the relative density of bimodal powder was significantly higher than the density of single-mode powder at lower power levels, but at higher power levels, density of single-mode powder exceeds the values of bimodal powder due to vaporization of finer particles in the bimodal mixture.

The trends displayed in Figure 4 implied that ultra-high energy densities were not necessarily needed for enhanced consolidation of bimodal powder. Using high power leads to a higher demand on energy resources. However, higher scan speeds can mean faster build efficiency than low scan speed. The combined effect of using high power and high scan speed to achieve a desired VED was found to be most effective on densification.

### 3.3. Porosity and Melt Pool

The cross-section of as polished SLM-manufactured samples parallel to the build direction was observed by optical microscopy to determine the area% porosity for samples produced at several VEDs. The parallel cross-section with respect to build direction of several parts made from GH single-mode and BIMO powder, and SLM-manufactured at power level of 203 W was observed at magnification of 50×. The optical micrographs of high-density samples (>99%) from single-mode and bimodal powder SLM-manufactured at a VED of 101.5 J/mm$^3$ are presented in Figure 5a,b, respectively. Image J software was used to analyze the optical micrographs and quantify porosity. The area percent porosity of each of these sections was averaged to give an approximate % porosity for samples produced from each powder type as presented in Table 3.

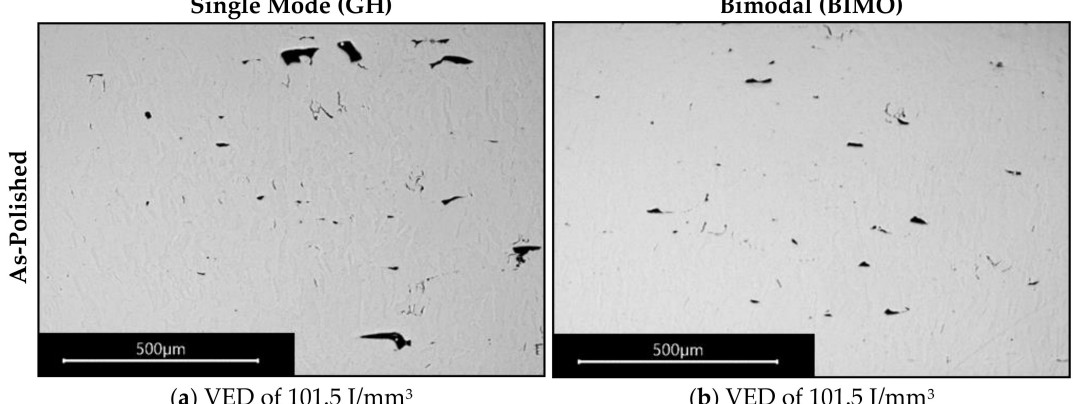

(**a**) VED of 101.5 J/mm³        (**b**) VED of 101.5 J/mm³

**Figure 5.** *Cont.*

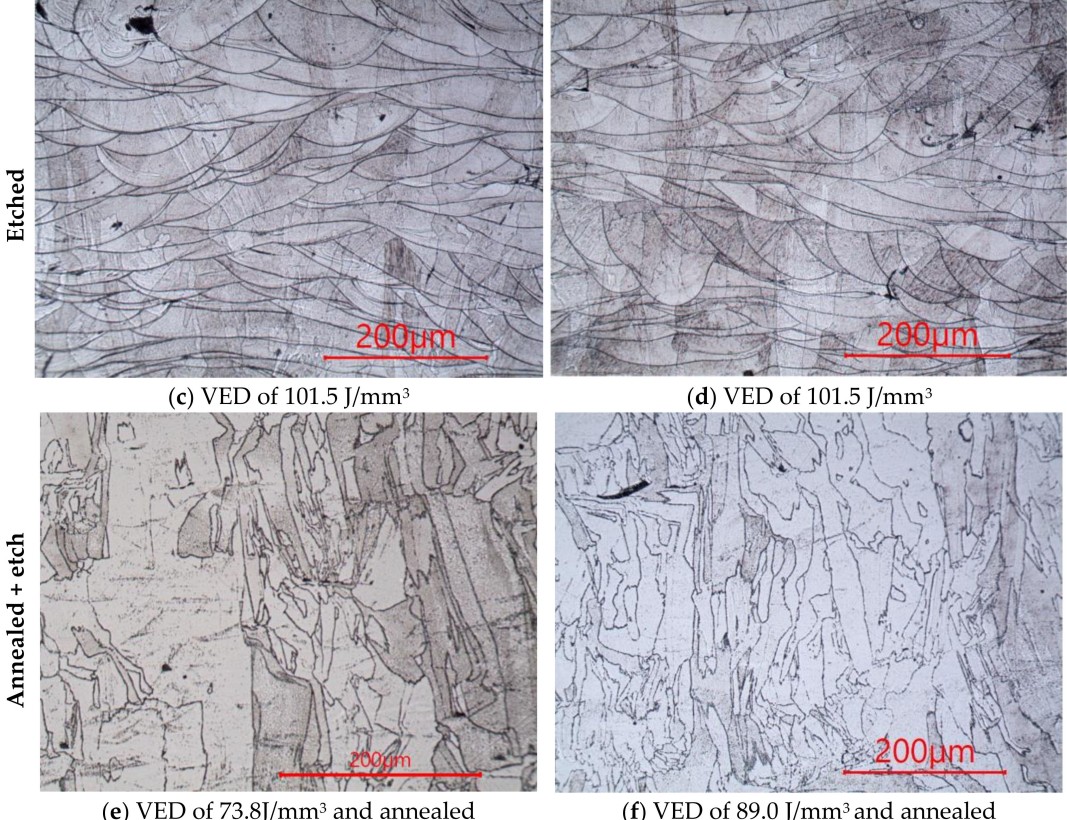

(**c**) VED of 101.5 J/mm³        (**d**) VED of 101.5 J/mm³

(**e**) VED of 73.8J/mm³ and annealed     (**f**) VED of 89.0 J/mm³ and annealed

**Figure 5.** (**a**,**b**) As-polished and (**c**,**d**) etched optical micrographs obtained from parallel cross-sections of (**a**,**c**) GH single-mode powder, and (**b**,**d**) BIMO bimodal powder, each SLM-manufactured at a VED of 101.5 J/mm³; (**e**) Annealed and etched GH sample SLM-manufactured at 73.8 J/mm³; (**f**) Annealed and etched BIMO sample SLM-manufactured at 89.0 J/mm³.

**Table 3.** Density values measured from area percent porosity based on analysis of optical micrographs of samples cut parallel to build direction and density values measured from the Archimedes method for parts made from single-mode GH powder and bimodal powder BIMO (parts were SLM-manufactured at VED of 90.2, 101.5 and 116 J/mm³).

| VED (J/mm³) | GH | | | BIMO | | |
|---|---|---|---|---|---|---|
| | Area Density | Archimedes Relative Density | Difference | Area Density | Archimedes Relative Density | Difference |
| 116.0 | 98.1 ± 1.2% | 99.1 ± 0.2% | −1.0% | 99.8 ± 0.1% | 98.7 ± 0.3% | 1.1% |
| 101.5 | 98.8 ± 0.2% | 99.3 ± 0.1% | −0.5% | 99.4 ± 0.1% | 99.2 ± 0.2% | 0.2% |
| 90.2 | 98.7 ± 0.1% | 98.3 ± 0.5% | 0.4% | 99.5 ± 0.1% | 99.1 ± 0.3% | 0.4% |

Since porosity was found to be most abundant between layers, only the parallel cross-section was used to obtain data presented in Table 3. While a single cross-section may not consistently provide an accurate view of the overall porosity of the sample, measured porosity using optical micrographs aligned closely with measured densities in this case, as shown in Table 3. The largest disparity was 1.06% density.

The height of the melt pool refers to the distance between the deepest level of penetration by a single melt track, and the deepest point of the next subsequent layer. The average melt pool height was determined by measuring the distance between the bottom edge of one melt pool to the bottom edge of a melt pool four layers above and dividing that distance by four. This was done because the

shift angle of the scanning pattern in these builds was 45°, and the scanning direction was repeated every 4 layers. A zig-zag scanning strategy was used with a shift angle of 45° between each layer.

Figure 5a–d shows optical micrographs of samples from single-mode and bimodal size distribution SLM-manufactured at VEDs of 101.5 J/mm$^3$. Optical micrographs were obtained from cross-section parallel to build direction.

Annealed and etched GH sample SLM-manufactured at 73.8 J/mm$^3$ and BIMO sample SLM-manufactured at 89.0 J/mm$^3$ are shown in Figure 5e,d, respectively. Figure 5c,d show the microstructure of SLM-manufactured samples with columnar grains and melt pool boundaries. Columnar grains in as-melted samples tended to grow preferentially perpendicular to the melt pool boundaries as shown and these growth directions were primarily maintained in the annealed samples as shown in Figure 5e,f. However, melt pool boundaries appear to have vanished after annealing. Some recrystallization was observed in the micrograph shown in Figure 5e,f. Although the melt boundaries were no longer visible in the annealed samples, phantom melt lines can be seen by following the pattern of smaller, more equiaxed grains along paths reminiscent of the curved cross-section of the melt pools. High diffusivity paths along grain boundaries and melt pool boundaries and high dislocation density associated with the SLM process, could theoretically facilitate recrystallization of grains. However, due to the absence of the appearance of annealing twins in Figure 5e,f, it appears that the annealing conditions were not sufficient to initiate new grain growth [33].

Table 4 presents the average thickness of melted layer for GH single-mode and BIMO bimodal powder size distribution at different levels of VEDs. The results in Table 4 show that average melt pool size did not vary significantly across this range of parameters. It is worth noting, however, that certain areas of porosity appear to result in inconsistent depth of melting, making the average layer height a rough approximation. Nonetheless, the layer height is shown to be, as anticipated, close to the machine-set layer height of 50 μm, but slightly lower due to volume lost in consolidation of each powder layer. Interestingly, the layer height of the bimodal powders shows higher values, suggesting that minimal consolidation from spread thickness occurred after melting. Greater layer thickness in bimodal powder was perhaps due to having a packed density of bimodal powder and better thermal conductivity in the as-built layer that likely led to a higher cooling rate in the as-built layer and a narrower melt pool with slightly increased layer height.

**Table 4.** Average thickness of melted layer for GH single-mode and BIMO powder at VED of 90.2, 101.5 and 116 J/mm$^3$.

| VED (J/mm$^3$) | Average Thickness of Melted Layer (μm) | |
| --- | --- | --- |
| | **GH Single Mode** | **BIMO** |
| **116.0** | 47.6 ± 2.8 | 49.7 ± 8.1 |
| **101.5** | 40.7± 6.0 | 49.2 ± 8.7 |
| **90.2** | 44.2 ± 11.0 | 57.3 ± 2.0 |

### 3.4. Sub-Grain Cellular Structure

SEM micrographs obtained from the microstructure of high-density samples made from single-mode and bimodal powder size distribution with VED of 81.2 and 116 J/mm$^3$ are shown in Figure 6. Corresponding cellular microstructure is shown at higher magnification in Figure 6a,b and Figure 6e,f, respectively. After collecting micrographs, Image J analysis was used to measure cell area and cell width. Based on sample regions shown in Figure 6, cells area and width were measured as presented in Table 5. In the case of single-mode powder size distribution, no significant difference was observed between the cell area of 0.424 ± 0.075 μm$^2$ associated with lower VED levels of 81.2 J/mm$^3$, and the cell area of 0.394 ± 0.061 μm$^2$ associated with higher VED levels of 116 J/mm$^3$. However, cell area for bimodal powder size distribution were measured to be 0.531 ± 0.065 μm$^2$ at lower VED of 81.2 J/mm$^3$ which was larger than 0.413 ± 0.019 μm$^2$ at higher VED level of 116 J/mm$^3$. Wang et al. [34] reported that lower energy density led to smaller intergranular cells in SLMed 316L, with an

average cell width of 0.31 μm at 104 J/mm$^3$ and 0.74 μm at a high VED of 179 J/mm$^3$. This suggests that while VED must be sufficiently high to achieve fully dense components, high VED values above ~125 J/mm$^3$ may be detrimental to mechanical properties. The reason for observing larger cells at lower VED for bimodal powder size distribution is not clear but could be likely due to high power of 203 W and possible vaporization of fine particles. Furthermore, cell area and width were larger for bimodal powder size distribution than in single-mode powder samples at both VED levels. A more extensive investigation is needed to identify the formation and growth mechanisms of these cellular substructures and how they may be affected by the PSD of feedstock powder used in SLM.

**Table 5.** Measured cell area and cell width of cellular structure observed in single-mode and bimodal powder size distribution SLM-manufactured at VED of 81.2 and 116 J/mm$^3$.

|  | Unit | GH Single Mode | | BIMO Bimodal | |
|---|---|---|---|---|---|
| **VED** | J/mm$^3$ | **116.0** | **81.2** | **116.0** | **81.2** |
| Cell area | μm$^2$ | 0.394 ± 0.061 | 0.424 ± 0.074 | 0.413 ± 0.019 | 0.531 ± 0.065 |
| Cell width | μm | 0.611 ± 0.037 | 0.636 ± 0.051 | 0.685 ± 0.063 | 0.774 ± 0.077 |

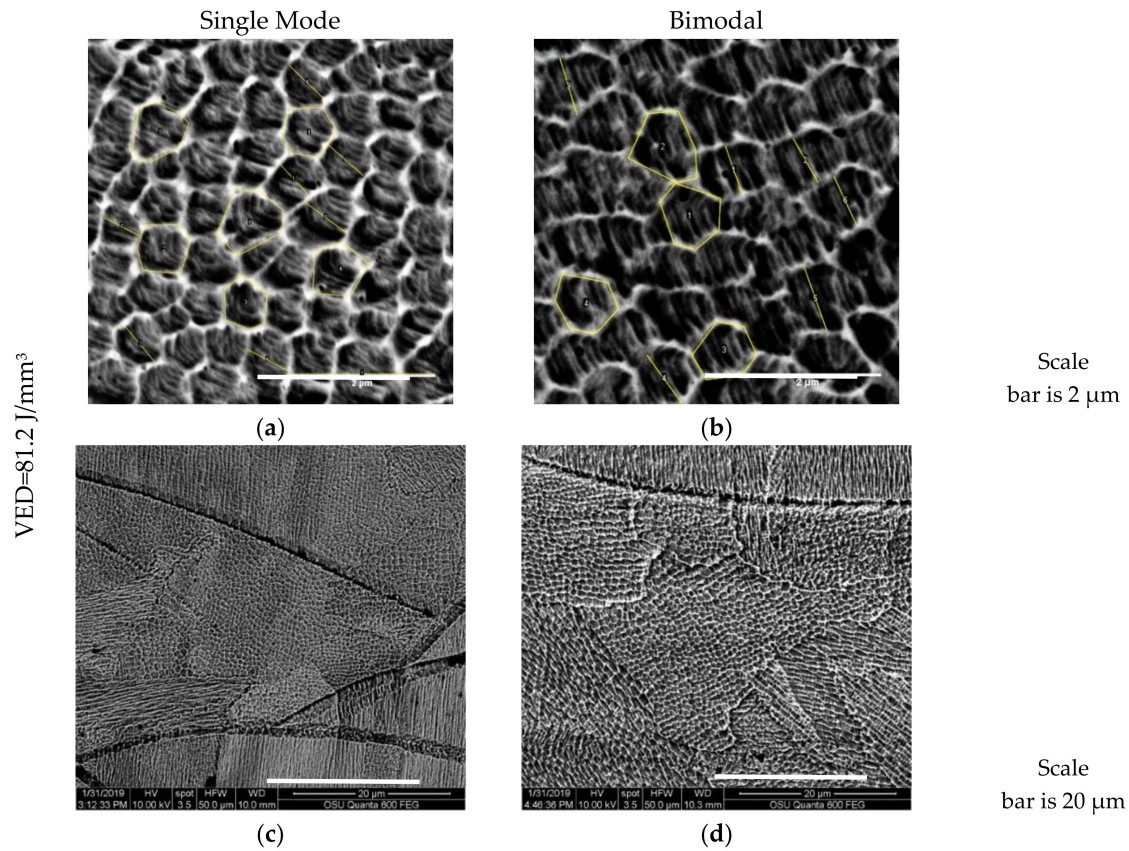

**Figure 6.** *Cont*.

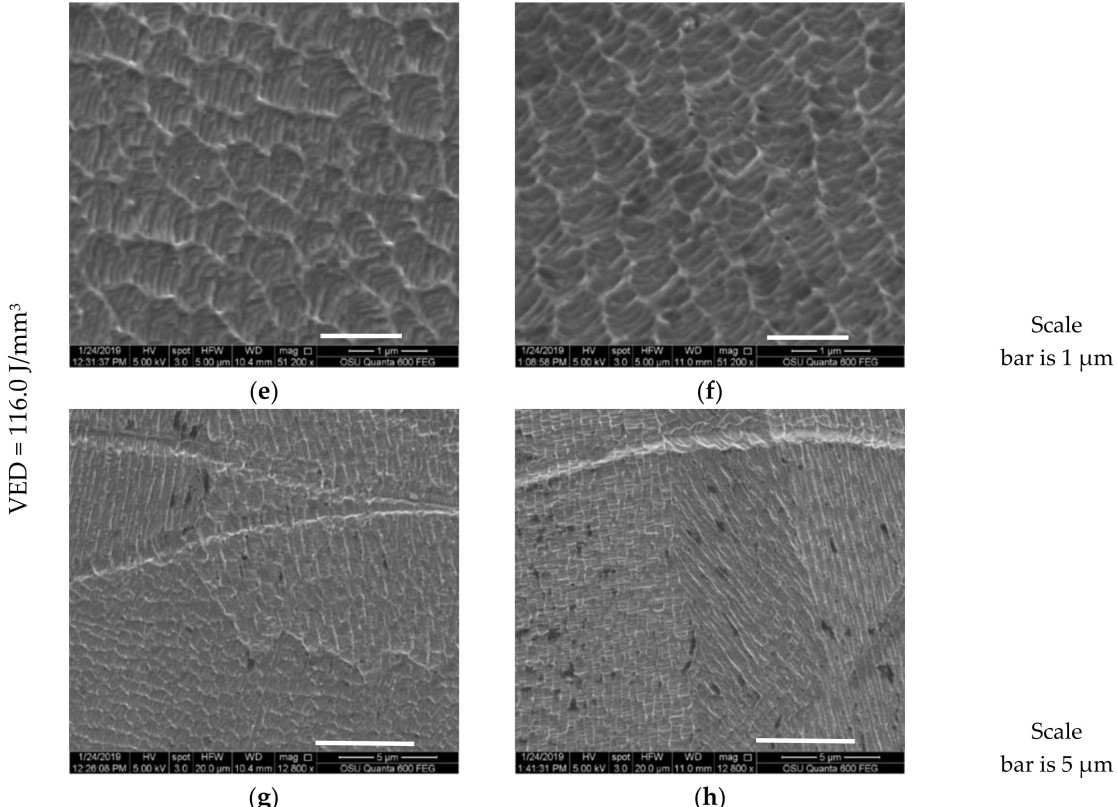

**Figure 6.** SEM micrographs obtained from (**a**,**c**) single-mode powder and (**b**,**d**) bimodal SLM-manufactured at 203 W, 1000 mm/s, VED = 81.2 J/mm$^3$; SEM micrograph obtained from single-mode (**e**,**g**) and bimodal (**f**,**h**) SLM-manufactured at 203 W, 700 mm/s, VED = 116.0 J/mm$^3$.

### 3.5. Microhardness

The averages of microhardness values at different VED values ranging from 66 to 116 J/mm$^3$ are shown in Figure 7. As shown in Figure 7, bimodal powder SLM-manufactured samples showed overall slightly higher hardness than single-mode powder SLM-manufactured samples. For example, at VED of 101 J/mm$^3$, microhardness values of 246 ± 6 HV and 239 ± 7 HV were measured for bimodal powder (BIMO) and single-mode (GH) powder, respectively. The average microhardness values for single-mode powder (GH) and bimodal powder (BIMO) measured in this study remained relatively constant with respect to VED for each powder batch (within standard deviation of each other) from 67.7 to 166.0 J/mm$^3$. Figure 7 suggests a tendency towards increasing hardness and decreasing density for a VED of 67.7 to 101.8 J/mm$^3$. Previous studies show a similar tendency towards increasing microhardness with increasing VED over a range of 50 to 80 J/mm$^3$ [29,34], though Wang et al. [34] found that beyond a VED of 125 J/mm$^3$, microhardness begins to decrease with increasing VED, due to the coarsening of the cellular structured in the SLMed material at higher energy levels. The microhardness values shown in Figure 7 were lower than the results from prior work by Ma et al. [17] who reported hardness of ~261 at VED of 68 J/mm$^3$. According to Ma et al. [17] hardness tends to decrease with increasing surface energy density, but not necessarily with VED. However, hardness measured in this study tended to be higher than Tucho et al.'s [18] reported 179–213 HV at 50–125 J/mm$^3$ and the 225 HV reported by Cherry et al. [19] for maximum-density samples produced at 104 J/mm$^3$.

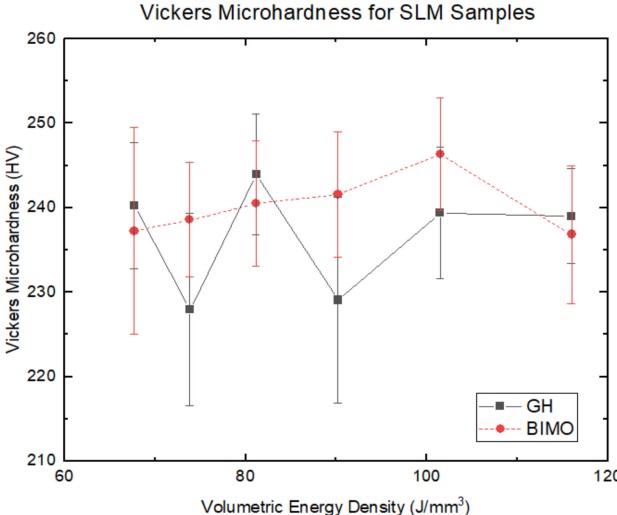

**Figure 7.** Vickers microhardness of SLM-manufactured samples produced from different powders at various VEDs in the highest power range (67.7–116.0 J/mm$^3$ at 203 W).

The XRD patterns of powder feedstock and SLM-manufactured parts were analyzed to identify the phase(s) in powder and parts and investigate any phase transformation occurred during the SLM process. Scans of as-built samples were taken from GH powder SLM-manufactured at VED of 67.7 J/mm$^3$ and BIMO powder SLM-manufactured at VED of 90.2 J/mm$^3$. Figure 8 shows the XRD patterns for each powder type, as well as the pattern for samples produced with each powder via SLM. All samples demonstrated patterns consistent with a pure austenite phase. Similarly, XRD patterns obtained from annealed GH sample, SLM-manufactured at 89.0 J/mm$^3$ and annealed BIMO SLM-manufactured at 73.8 J/mm$^3$ revealed the presence of pure austenite as shown in Figure 8.

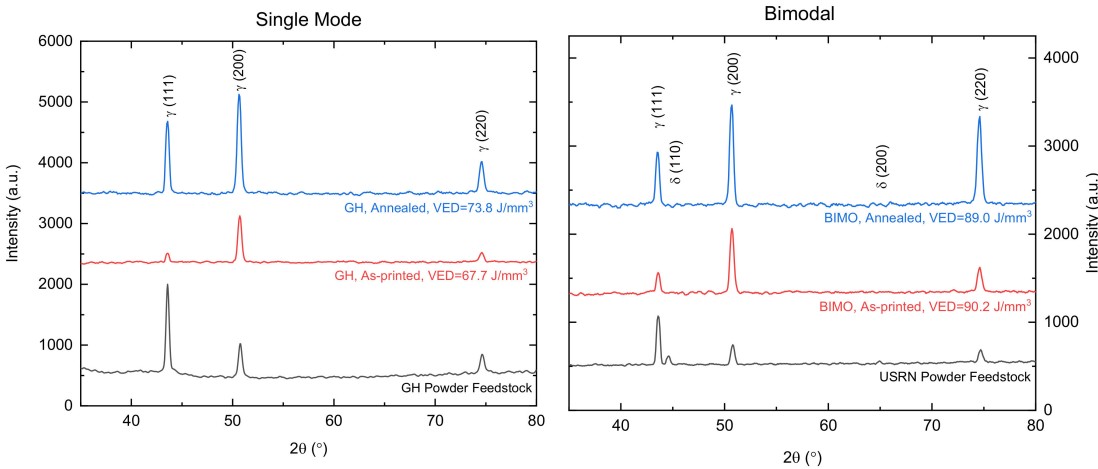

**Figure 8.** XRD pattern of GH large powder and BIMO bimodal powder, along with patterns of as-printed and annealed SLM-manufactured samples produced from each powder type at different VEDs, showing a primarily austenite phase for each.

Identified peaks of (111), (200) and (220) were attributed to the presence of γ-austenite phase in GH and SLM-manufactured samples produced from bimodal powder size distribution with no additional phases observed. In the BIMO powder feedstock, containing USRN small powder, several low-intensity peaks occurring at angles consistent with a δ-ferrite phase were present. This may be due to the fast quenching of molten material occurring during the manufacture of water-atomized powder found in the USRN feedstock. Similar evidence of ferrite was seen in XRD scans of 316L SLM feedstock powder by Sun et al. and Kurzynowski et al. [7,9]. However, in all cases, these peaks disappeared after SLM.

The intensity of peaks varied between GH powder feedstock and SLM-manufactured parts. XRD scans of powder consistently had (111) as the highest intensity peak, whereas (200) peaks had the highest intensity in the latter. This variation in peak intensity could reveal useful information about crystallographic orientation and developed texture in SLM-manufactured parts that will require additional characterization by electron back scattered diffraction (EBSD) and could be a potentially significant topic for our future work.

*3.6. Limitations and Future Work*

Inherent limitations in the scope and assembly used in this study leave room for further investigation to validate current findings and explore other aspects of performing SLM with bimodal feedstock powder. The unique rotational spreading motion of the SLM system used in this study resulted in an inconsistent powder coating velocity between different locations on the build platform relative to the center of rotation. Further study may require investigation of the effect of coater speed on the density of the powder bed. This may be particularly relevant to powders with less than excellent flow properties, such as the bimodal feedstock used in this study. Furthermore, a wider set of power and scan speed for SLM-manufactured parts should be explored to investigate the density of parts produced at lower VEDs with higher power and scan speed. In the SLM system used in this study, the power level can only reach approximately 225 W, but certainly the limits of the upward trajectory of the 100 W (50% power) curve could be tested by running at extremely slow scan speeds. Single-track experiments could be useful in the future work to quantify the effect of energy parameters on the melt pool geometry for both single-mode and bimodal powder size distribution.

## 4. Conclusions

The role of powder feedstock particle size distribution on density, microstructure and mechanical properties of 316L stainless steel components additively manufactured via the SLM process was investigated by varying energy densities from 35.7 to 116.0 $J/mm^3$ using both single-mode and bimodal particle size distribution. In this investigation, the following conclusions were made:

1.  Powder beds with bimodal size distributions could have higher maximum packing densities than normally distributed powders in a similar size range. The tap density of the bimodal mixture measured in this study was over 2% higher on average than the single-mode precursor powders.
2.  Bimodal powder size distribution showed poor flow characteristics compared to single-mode powder. Low apparent density in small and bimodal powders resulted in higher Hausner ratios. Flowability was diminished when using bimodal powders in this size range. Use of spherical small particles in a bimodal powder distribution, under a low oxygen atmosphere, might have a positive effect on bulk printed samples, but moisture absorption and flow challenges in small particles of irregular shape (high friction and surface area) made them practically less useful for SLM experiments.
3.  Optimum VED for achieving high-density SLM-manufactured parts does not exist as a single value for a given powder type, and rather dynamically changes with laser power. The relationship between VED and relative material density plateaus at lower power input levels for bimodal powders than for single-mode powders. Below ~200 W, with the constant parameters used in this study, bimodal powder had consistently higher as-built density than single-mode powder. Beyond 200 W, increasing VED (81.2 $J/mm^3$) decreased material density in bimodal powders, possibly due to the small powder in the mixture vaporizing with higher energy input, leaving voids from vapor recoil pressure in the larger powder left behind.
4.  At VED of 101 $J/mm^3$, SLM-manufactured samples from bimodal powders showed hardness of 246 ± 6 HV whereas single-mode powders showed hardness of 239 ± 7 HV. SLM-manufactured samples from both powders yielded higher microhardness values than reported values for wrought 316L (~200 HV).

5.  Melt pool depth is relatively consistent between single-mode and bimodal samples at the same VED. Microstructure shows no discernable differences related to powder type used. Annealing resulted in coarsening of grains within the melt pool and formation of new grains along melt pool boundaries similar for both bimodal and single-mode powder.

**Author Contributions:** Conceptualization, methodology, and verification for this project were performed as a collaboration between H.C. and S.P. Formal analysis, investigation, data curation, visualization, and writing—original draft preparation were contributed by H.C. Resources, supervision, project administration, funding acquisition, and writing—review and editing were provided by S.P. All authors have read and agreed to the published version of the manuscript.

**Funding:** This research was made possible by the financial support of Oregon Metal Initiative (OMI) and Oregon Manufacturing Innovation Center (OMIC); as well as Nicholas Cunningham and Noah Philips at ATI Specialty alloys.

**Acknowledgments:** The authors wish to thank the staff and facility personnel at ATAMI for providing resources and facilities that supported this work. Further, the authors express gratitude to Pete Eschbach and Teresa Sawyer at the OSU Electron Microscopy Center and the assistance of Katrina Singh and James Valencia through the Oregon State University Summer Experience in Science and Engineering for Youth program (SESEY).

**Conflicts of Interest:** The authors declare no conflict of interest.

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
