# Peer review of "Use of Bimodal Particle Size Distribution in Selective Laser Melting of 316L Stainless Steel"

_jmmp, doi:10.3390/jmmp4010008_

Round 1

Reviewer 1 Report

Dear Authors,

Congratulations on your work, which is about a hot topic, being well organized and described. However, there are some issues I would like to see clarified or amended, such as:

Main concerns:

There is no contextualization about the selection of the AISI 316L material for this work. The sentence in line 41: "...one two-dimensional layer at a time ..." is not correct, because the thickness plays an important role in the 3D printing process. Please correct this sentence. In the variables used in Equation 1, please include the units for each variable. In line 174, the particle size is displayed, but no standard deviation is pointed out. It is a crucial question in this kind of works. Moreover, a graph with the particles size distribution would be welcome.In lines 253-254 you talk about the annealing process. However, th information is very scarce. How did you select 2 hours (because it depends on the thickness of the specimen to be treated)? How did you select 1020ºC? How did you select 8.5ºC/min for the heating rate? You need to explain all your options. Figure 2 presents a high dispersion of the particles size. You need to complement these pictures with the particles distribution size diagram. Observing Figure 5 (c and d), it seems that the width of the different layers is not regular. Thus, you need to comment this. In lines 358 to 360, you need to justify the sentence: "Area% porosity can provide an even better representation...". The following sentence is more accurate, but the first one can induce doubts in the reader. The difference between grains size, in some cases, is intriguing. Could you explain better what is going on regarding these differences? How can we ensure the mechanical behaviour of the material with these differences? Is the material isotropic? 

Minor concerns:

Please use the same number of significant digits when you use different values in each table. It is a scientific concern. You use randomly space or not between values and units. Please use always a space between units and values. The management of spaces need to be cared because there are sites where there are more than one space between words. Please correct this (lines, 157, 158, 186, 248, etc.) In line 177, please put Metals with capital letter. In line 210, please explain what is "...5 x 5 mm cylindrical specimens". It seems that the diameter symbol is missing. In line 231, please explain the "DI" meaning (extended name - Deionised ?)

Good luck. Hope this review can help you improving the paper.

Kind regards,

FGS

Author Response

Response to Reviewer 1:

We really appreciate this comment and addressed all the questions accordingly.

There is no contextualization about the selection of the AISI 316L material for this work. We selected the AISI 316L because it is the most commonly used stainless steel in industry with a wide range of applications, and also the most common steel that has been processed by SLM. This part has been added to the manuscript

In Manuscript: The AISI 316L is the most commonly used stainless steel in industry due to its strength, toughness, and corrosion resistance [2] and therefore, 316L is a desirable system for use in SLM and this study.

Use of bimodal particle size distribution was invariant from the alloy, so we chose 316L as a workhorse of industry, but the results can be extended beyond just 316L and can be used for other alloy systems.

The sentence in line 41: "...one two-dimensional layer at a time ..." is not correct, because the thickness plays an important role in the 3D printing process. Please correct this sentence.

It is removed, and the sentence is now corrected:

In Manuscript: Selective laser melting (SLM) is a laser powder bed fusion (LPBF) additive manufacturing (AM) process in which three-dimensional parts are manufactured by scanning a laser in prescribed pattern on the surface of a bed of metal powder, melting the material which rapidly solidifies, before a new layer of powder is spread to repeat the process.

In the variables used in Equation 1, please include the units for each variable.

It is addressed accordingly.

In Manuscript: The intensity of a laser beam as it melts metal powder during SLM is dependent on parameters such as power (P) [W], scan speed (v) [mm/s], layer thickness (t) [µm], and laser beam diameter (σ) [µm].

In line 174, the particle size is displayed, but no standard deviation is pointed out. It is a crucial question in this kind of works. Moreover, a graph with the particles size distribution would be welcome.

The standard deviation is added. We tried using the Malvern particle size analyzer but due to the fine size of particles we could not get repeatable meaningful results.

In Manuscript: D50 = 36.31±11.92 µm; and a smaller, semi-spherical water-atomized and gas-atomized powder from US Research Nanomaterials (USRN) with a primary particle size of D50 = 5.52±2.5 µm.

D10= 2.71+/- 0.324 um, 

D50= 5.56+/- 0.181 um, 

D90= 11.5 +/- 0.476 um

The particle size distribution provided by the vendor is shown above.

In lines 253-254 you talk about the annealing process. However, th information is very scarce. How did you select 2 hours (because it depends on the thickness of the specimen to be treated)? How did you select 1020ºC? How did you select 8.5ºC/min for the heating rate? You need to explain all your options.

We chose 1020 °C and 2 hrs to make sure we get fully austenitic phase. The ramp was a standard ramp in our box furnace. This heat treatment has also been used by other researchers, for example Salman et al. [Materials Science and Engineering: A, Volume 748, 4 March 2019, Pages 205-212] studied the effect of heat treatment on microstructure and mechanical properties of 316L steel synthesized by selective laser melting. They used 1273K which is very similar to 1020 °C we used here.

In manuscript: Selected samples were annealed by heating at a ramp rate of 8.5°C/min and holding for 2hrs at 1020°C under a nitrogen atmosphere to ensure a fully austenitic phase was achieved.   

Figure 2 presents a high dispersion of the particles size. You need to complement these pictures with the particles distribution size diagram.

Respectfully, we think the distribution in the larger powder (Fig. 2b) is not that significant and it is within the standard deviation range and the accepted particle size distribution range for SLM process. However, we agree that distribution in smaller powder seems a bit large. We are replacing it with a better SEM micrograph. As you are aware, SEM captures a small area of specimen and distribution variation is an artifact. For the Fig. 2c, there should be a large distribution as it is the goal of this study.

As explained in comment 4, we tried using the Malvern particle size analyzer but due to the fine size of particles we could not get repeatable meaningful results. We replaced the Figure 2a with a better micrograph.

D10= 2.71+/- 0.324 um, 

D50= 5.56+/- 0.181 um, 

D90= 11.5 +/- 0.476 um

The particle size distribution provided by the vendor is shown above.

Observing Figure 5 (c and d), it seems that the width of the different layers is not regular. Thus, you need to comment this.

We agree. In Fig. 5c and Fig. 5d, the average thickness of melted layer for 101. 5/mm3, was 40.7± 6.01 µm for single mode and 49.2 ± 8.70 µm for bimodal powder, respectively. We think greater thickness in bimodal powder was perhaps due to having a packed density of bimodal powder and better thermal conductivity in the as-built layer  that leads to a higher cooling rate in the as-built layer and an increase the layer height but the overall melt pool size became narrower.

Tan et al. [Journal of Alloys and Compounds, Volume 787, 30 May 2019, Pages 903-908] has explained that high thermal conductivity of the as-fabricated layers lead to that the columnar grains in the rods oriented to the building direction and longitudinal direction of the rod respectively. In addition, the length of grains in the nodes was smaller than that in the rods, which was attributed to the smaller size molten pool caused by greater cooling rate. As a consequence, the hardness of the node was higher than that of the rod.

We also observed the same increase in hardness similar to Tan et al work.

In manuscript: The layer height of the bimodal powders shows higher values, suggesting that minimal consolidation from spread thickness occurred after melting. Greater layer thickness in bimodal powder was perhaps due to having a packed density of bimodal powder and better thermal conductivity in the as-built layer  that likely led to a higher cooling rate in the as-built layer and a narrower melt pool with slightly increased layer height.

In lines 358 to 360, you need to justify the sentence: "Area% porosity can provide an even better representation...". The following sentence is more accurate, but the first one can induce doubts in the reader.

We agree. This is removed and, the sentence is revised accordingly.

In manuscript: Since porosity was found to be most abundant between layers, only the parallel cross-section was used to obtain data presented in Table 3. While a single cross-section may not consistently provide an accurate view of the overall porosity of the sample, measured porosity using optical micrographs aligned closely with measured densities in this case, as shown in Table 3. The largest disparity was 1.06% density.

The difference between grains size, in some cases, is intriguing. Could you explain better what is going on regarding these differences?

Truthfully, we do not know why there may be differences in grain size between particular feedstock PSDs. That would likely be a whole additional chapter.

How can we ensure the mechanical behavior of the material with these differences? Is the material isotropic?

We measured the microhardness on the cross section perpendicular to the build direction. We did not build multiple parts with different orientations in order to measure tensile properties. This was not the topic of this study. Therefore, we do not have data on the isotropy of the samples. It can be a topic of our future study.

Please use the same number of significant digits when you use different values in each table. It is a scientific concern.

It is addressed accordingly.  

You use randomly space or not between values and units. Please use always a space between units and values.

It is addressed accordingly.

The management of spaces need to be cared because there are sites where there are more than one space between words. Please correct this (lines, 157, 158, 186, 248, etc.)

It is addressed accordingly.

In line 177, please put Metals with capital letter.

It is addressed accordingly.

In line 210, please explain what is "...5 x 5 mm cylindrical specimens". It seems that the diameter symbol is missing.

It is addressed accordingly.

In line 231, please explain the "DI" meaning (extended name - Deionised ?)

It is addressed accordingly.

Reviewer 2 Report

Paper is delighted with build quality of 3D printed 316L with 2 powder diameter sizes. It is interesting, but some corrections are needed.

Abstract is way too long and should be shortened to about half of current length. Similar holds for introduction, which reads like thesis and not scientific paper. Limit it to the most important things and shorten it. Stated C content in introduction is wrong, as 0.03 wt% is correct number not 0.3. Please add which % is meant where, wt, vol., at.... In experimental methods mL should be converted to cm3. Please add powder SEM sample preparation technique. Similar in section 2.2, hardness measurement weight and dwell time are missing. Dwell time for XRD is also not given. Please explain why smaller diameter powder was not used separately for experiments and compared to large and bimodal ones. In results Table 3 is very limited compared to figure 4. Is there reason only some VED values are compared to area and volume densities? Regarding section 3.6 are presented results repeatable?

Author Response

Response to Reviewer 2:

Abstract is way too long and should be shortened to about half of current length.

It is addressed accordingly

Abstract: Spherical powders with single mode (D50 = 36.31 µm), and bimodal (D50,L = 36.31 µm, D50,s = 5.52 µm) PSD were used in selective laser melting of 316L stainless steel in nitrogen atmosphere at volumetric energy densities (VED) ranging from 35.7-116.0 J/mm3. Bimodal particle size distribution could provide up to 2% greater tap density than single mode powder. For low laser power (107-178 W) where relative density was <99%, bimodal feedstock resulted in higher density than single mode feedstock. However, at higher power (>203 W), the density of bimodal-fed components decreased as the VED increased, likely due to vaporizing of the fine powder in bimodal distributions. Size of intergranular cell regions did not appear to vary significantly between single mode and bimodal specimens (0.394-0.531 µm2 at 81-116 J/mm3). Despite higher packing densities in powder feedstock with bimodal PSD, the results of this study suggest that differences in conduction melting and vaporization points between the two primary particle sizes would limit the maximum achievable density of SLMed components produced from bimodal powder.

Similar holds for introduction, which reads like thesis and not scientific paper. Limit it to the most important things and shorten it.

It is addressed accordingly. We reduced it from 4 pages to 2 pages total introduction.

Stated C content in introduction is wrong, as 0.03 wt% is correct number not 0.3.

It is addressed accordingly and now is corrected.

Please add which % is meant where, wt, vol., at.... In experimental methods mL should be converted to cm3.

It is addressed accordingly.

Please add powder SEM sample preparation technique.

This is added.

Each SEM powder sample was prepared by applying a small amount of powder to carbon tape adhesive on the sample’s holder, then using sweeping away excess loose powder using a hose of flowing nitrogen gas.

Similar in section 2.2, hardness measurement weight and dwell time are missing.

This is added.  We used 500 g force and 13 seconds dwell time.

In Manuscript: Leco LM 248AT Vickers microhardness tester with three samples taken from the normal and build plane of each specimen. Loading was at done at 500 g with a dwell time of 13 seconds. For each powder type and parameter set measured, 10 indentations were made to record the average hardness and standard deviation.

Dwell time for XRD is also not given.

This is added.

Continuous scanning mode was used to conduct scans at 2°/min with a dwell time of 1.5s for a range of 2θ = 20-90° at step size of 0.05o.

Please explain why smaller diameter powder was not used separately for experiments and compared to large and bimodal ones.

The reason for this was bad powder flow and spreading properties in fine powder. It did not give a high density powder bed to begin with, and high density parts could not be printed.

In results Table 3 is very limited compared to figure 4. Is there reason only some VED values are compared to area and volume densities?

The VED selected for Table 3 had the highest densities that were in a small range (from 98.1-99.8%). If the density was lower than this range , the Archimedes method was not effective and accurate in measuring the density because pores were filled with water, and inaccurate density values were measured.  

Regarding section 3.6 are presented results repeatable? 

Yes. As long as the powder particle size distribution and power and scanning speed are constant, results can be repeatable.

Reviewer 3 Report

The article presents interesting study about additive manufacturing of 316L steel. The comparison of properties of monosized and bimodalsized powder is of practical importance. I have several comments and remarks to the authors:

1) The strategy of scanning should be mentioned.

2) In the whole manuscript, there is a problem with statistic of presented data. The authors present density e.g. 5.23 +/- 0.111 g/cm3 - the number of digits should be the same for value and error. The same statistic problem is in evaluation of hardness data: "On average, SLM manufactured samples from bimodal powders showed slightly greater hardness (236.8 ± 8.1 HV) than those from single mode (236.6 ± 8.6 HV)." This sentence contains two problems - the first one is statistic, because the numbers (looking at errors) are equal. On the top, the Vickers hardness is not very precise method and the difference should be about 30 HV different.

3) The scalebars should be more visible (especially in Fig. 6).

4) There is also typografic problem: there are no spaces between number and unit.

Author Response

Response:

We appreciate comments of the reviewer 3.

The strategy of scanning should be mentioned.

It is added to the manuscript:

A zig-zag scanning strategy was used with a shift angle of 45° between each layer.   

In the whole manuscript, there is a problem with statistic of presented data. The authors present density e.g. 5.23 +/- 0.111 g/cm3 - the number of digits should be the same for value and error. The same statistic problem is in evaluation of hardness data: "On average, SLM manufactured samples from bimodal powders showed slightly greater hardness (236.8 ± 8.1 HV) than those from single mode (236.6 ± 8.6 HV)." This sentence contains two problems - the first one is statistic, because the numbers (looking at errors) are equal. On the top, the Vickers hardness is not very precise method and the difference should be about 30 HV different.

We agree and this is now corrected in the manuscript.

The scalebars should be more visible (especially in Fig. 6).

We agree and this is now corrected in the manuscript.

There is also typografic problem: there are no spaces between number and unit.

We agree and this is now corrected in the manuscript.

Round 2

Reviewer 2 Report

Authors made corrections according to comments given previously.